# Reparameterized Variational Divergence Minimization for Stable Imitation

## Abstract

State-of-the-art results in imitation learning are currently held by adversarial methods that iteratively estimate the divergence between student and expert policies and then minimize this divergence to bring the imitation policy closer to expert behavior. Analogous techniques for imitation learning from observations alone (without expert action labels), however, have not enjoyed the same ubiquitous successes. Recent work in adversarial methods for generative models has shown that the measure used to judge the discrepancy between real and synthetic samples is an algorithmic design choice, and that different choices can result in significant differences in model performance. Choices including Wasserstein distance and various $f$-divergences have already been explored in the adversarial networks literature, while more recently the latter class has been investigated for imitation learning (Ke et al., 2019). Unfortunately, we find that in practice this existing imitation-learning framework for using $f$-divergences suffers from numerical instabilities stemming from the combination of function approximation and policy-gradient reinforcement learning. In this work, we alleviate these challenges and offer a reparameterization of adversarial imitation learning as $f$-divergence minimization before further extending the framework to handle the problem of *imitation from observations only*. Empirically, we demonstrate that our design choices for coupling imitation learning and $f$-divergences are critical to recovering successful imitation policies. Moreover, we find that with the appropriate choice of $f$-divergence, we can obtain imitation-from-observation algorithms that outperform baseline approaches and more closely match expert performance in continous-control tasks with low-dimensional observation spaces. With high-dimensional observations, we still observe a significant gap with and without action labels, offering an interesting avenue for future work.

## 1 Introduction

Imitation Learning (IL) (Osa et al., 2018) refers to a paradigm of reinforcement learning in which the learning agent has access to an optimal, reward-maximizing expert for the underlying environment. In most work, this access is provided through a dataset of trajectories where each observed state is annotated with the action prescribed by the expert policy. This is often an extremely powerful learning paradigm in contrast to standard reinforcement learning, since not all tasks of interest admit easily-specified reward functions. Additionally, not all environments are amenable to the prolonged and potentially unsafe exploration needed for reward-maximizing agents to arrive at satisfactory policies (Achiam et al., 2017; Chow et al., 2019).

While the traditional formulation of the IL problem assumes access to optimal expert action labels, the provision of such information can often be laborious (in the case of a real, human expert) or incur significant financial cost (such as using elaborate instrumentation to record expert actions). Additionally, this restrictive assumption removes a vast number of rich, observation-only data sources from consideration (Zhou et al., 2018). To bypass these challenges, recent work (Liu et al., 2018; Torabi et al., 2018a;b; Edwards et al., 2019; Sun et al., 2019) has explored what is perhaps a more natural problem formulation in which an agent must recover an imitation policy from a dataset containing only expert observation sequences. While this Imitation Learning from Observations (ILfO) setting carries tremendous potential, such as enabling an agent to learn complex tasks from watching freely available videos on the Internet, it also is fraught with significant additional challenges. In

this paper, we show how to incorporate recent advances in generative-adversarial training of deep neural networks to tackle imitation-learning problems and advance the state-of-the-art in ILfO.

With these considerations in mind, the overarching goal of this work is to enable sample-efficient imitation from expert demonstrations, both with and without the provision of expert action labels.

The rich literature on Generative Adversarial Networks (Goodfellow et al., 2014) has expanded in recent years to include alternative formulations of the underlying objective that yield qualitatively different solutions to the saddle-point optimization problem (Li et al., 2015; Dziugaite et al., 2015; Zhao et al., 2016; Nowozin et al., 2016; Arjovsky et al., 2017; Gulrajani et al., 2017). Of notable interest are the findings of Nowozin et al. (2016) who present Variational Divergence Minimization (VDM), a generalization of the generative-adversarial approach to arbitrary choices of distance measures between probability distributions drawn from the class of $f$-divergences (Ali & Silvey, 1966; Csiszár et al., 2004). Applying VDM with varying choices of $f$-divergence, Nowozin et al. (2016) encounter learned synthetic distribu-

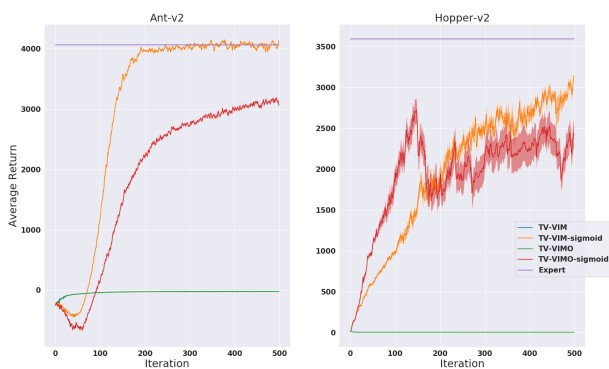

Figure 1: Evaluating the original $f$-VIM framework (and its ILfO counterpart, $f$-VIMO) in the Ant ($\mathcal{S} \in \mathbb{R}^{111}$) and Hopper ($\mathcal{S} \in \mathbb{R}^{11}$) domains with the Total Variation distance. $f$-VIM/VIMO-sigmoid denotes our instantiation of the frameworks, detailed in Sections 4.2 and 4.3. Note that, in both plots, the lines for TV-VIM and TV-VIMO overlap.

tions that can exhibit differences from one another while producing equally realistic samples. Translating this idea for imitation is complicated by the fact that the optimization of the generator occurs via policy-gradient reinforcement learning (Sutton et al., 2000). Existing work in combining adversarial IL and $f$-divergences (Ke et al., 2019), despite being well-motivated, fails to account for this difference; the end results (shown partially in Figure 1, where TV-VIM is the method of Ke et al. (2019), and discussed further in later sections) are imitation-learning algorithms that scale poorly to environments with higher-dimensional observations.

In this work, we assess the effect of the VDM principle and consideration of alternative $f$-divergences in the contexts of IL and ILfO. We begin by reparameterizing the framework of Ke et al. (2019) for the standard IL problem. Our version transparently exposes the choices practitioners must make when designing adversarial imitation algorithms for arbitrary choices of $f$-divergence. We then offer a single instantiation of our framework that, in practice, allows stable training of good policies across multiple choices of $f$-divergence. An example is illustrated in Figure 1 where our methods (TV-VIM-sigmoid and TV-VIMO-sigmoid) result in significantly superior policies. We go on to extend our framework to encapsulate the ILfO setting and examine the efficacy of the resulting new algorithms across a range of continuous-control tasks in the MuJoCo (Todorov et al., 2012) domain. Our empirical results validate our framework as a viable unification of adversarial imitation methods under the VDM principle. With the assistance of recent advances in stabilizing regularization for adversarial training (Mescheder et al., 2018), improvements in performance can be attained under an appropriate choice of $f$-divergence. However, there is still a significant performance gap between the recovered imitation policies and expert behavior for tasks with high dimensional observations, leaving open directions for future work in developing improved ILfO algorithms.

## 2    RELATED WORK

The algorithms presented in this work fall in with inverse reinforcement learning (IRL) (Ng et al.; Abbeel & Ng, 2004; Syed & Schapire, 2007; Ziebart et al., 2008; Finn et al., 2016; Ho & Ermon, 2016) approaches to IL. Early successes in this regime tend to rely on hand-engineered feature rep-

resentations for success (Abbeel & Ng, 2004; Ziebart et al., 2008; Levine et al., 2011). Only in recent years, with the aid of deep neural networks, has there been a surge in the number of approaches that are capable of scaling to the raw, high-dimensional observations found in real-world control problems (Finn et al., 2016; Ho & Ermon, 2016; Duan et al., 2017; Li et al., 2017; Fu et al., 2017; Kim & Park, 2018). Our work focuses attention exclusively on adversarial methods for their widespread effectiveness across a range of imitation tasks without requiring interactive experts (Ho & Ermon, 2016; Li et al., 2017; Fu et al., 2017; Kostrikov et al., 2018); at the heart of these methods is the Generative Adversarial Imitation Learning (GAIL) (Ho & Ermon, 2016) approach which produces high-fidelity imitation policies and achieves state-of-the-art results across numerous continuous-control benchmarks by leveraging the expressive power of Generative Adversarial Networks (GANs) (Goodfellow et al., 2014) for modeling complex distributions over a high-dimensional support. From an IRL perspective, GAIL can be viewed as iteratively optimizing a parameterized reward function (discriminator) that, when used to optimize an imitation policy (generator) via policy-gradient reinforcement learning (Sutton et al., 2000), allows the agent to shift its own behavior closer to that of the expert. From the perspective of GANs, this is achieved by discriminating between the respective distributions over state-action pairs visited by the imitation and expert policies before training a generator to fool the discriminator and induce a state-action visitation distribution similar to that of the expert.

While a large body of prior work exists for IL, recent work has drawn attention to the more challenging problem of imitation learning from observation (Sermanet et al., 2017; Liu et al., 2018; Goo & Niekum, 2018; Kimura et al., 2018; Torabi et al., 2018a;b; Edwards et al., 2019; Sun et al., 2019). To more closely resemble observational learning in humans and leverage the wealth of publicly-available, observation-only data sources, the ILfO problem considers learning from expert demonstration data where no expert action labels are provided. Many early approaches to ILfO use expert observation sequences to learn a semantic embedding space so that distances between observation sequences of the imitation and expert policies can serve as a cost signal to be minimized via reinforcement learning (Gupta et al., 2017; Sermanet et al., 2017; Dwibedi et al., 2018; Liu et al., 2018). In contrast, Torabi et al. (2018a) introduce Behavioral Cloning from Observation (BCO) which leverages state-action trajectories collected under a random policy to train an inverse dynamics model for inferring the action responsible for a transition between two input states (assuming the two represent a state and next-state pair). With this inverse model in hand, the observation-only demonstration data can be converted into the more traditional dataset of state-action pairs over which standard BC can be applied. Recognizing the previously discussed limitations of BC approaches, Torabi et al. (2018b) introduce the natural GAIL counterpart for ILfO, Generative Adversarial Imitation from Observation (GAIFO); GAIFO is identical to GAIL except the distributions under consideration in the adversarial game are over state transitions (state and next-state pairs), as opposed to state-action pairs requiring expert action labels. While Torabi et al. (2018b) offer empirical results for continuous-control tasks with low-dimensional features as well as raw image observations, GAIFO falls short of expert performance in both settings leaving an open challenge for scalable ILfO algorithms that achieve expert performance across a wide spectrum of tasks. A central question of this work is to explore how alternative formulations of the GAN objective that underlies GAIFO might yield superior ILfO algorithms. For a more in-depth survey of ILfO approaches, we refer readers to Torabi et al. (2019). We refer readers to the Appendix for a broader overview of prior work.

## 3 BACKGROUND

We begin by formulating the problems of imitation learning and imitation learning from observation respectively before taking a closer look at $f$-divergences and connecting them to imitation learning.

### 3.1 IMITATION LEARNING & IMITATION FROM OBSERVATION

We operate within the Markov Decision Process (MDP) formalism (Bellman, 1957; Puterman, 2014) defined as a five-tuple $\mathcal{M} = \langle \mathcal{S}, \mathcal{A}, \mathcal{R}, \mathcal{T}, \gamma \rangle$ where $\mathcal{S}$ denotes a (potentially infinite) set of states, $\mathcal{A}$ denotes a (potentially infinite) set of actions, $\mathcal{R} : \mathcal{S} \times \mathcal{A} \times \mathcal{S} \to \mathbb{R}$ is a reward function, $\mathcal{T} : \mathcal{S} \times \mathcal{A} \to \Delta(\mathcal{S})$ is a transition function, and $\gamma \in [0, 1)$ is a discount factor. At each timestep, the agent observes the current state of the world, $s_t \in \mathcal{S}$, and randomly samples an action according to its stochastic policy $\pi : \mathcal{S} \to \Delta(\mathcal{A})$. The environment then transitions to a new state according to

the transition function $\mathcal{T}$ and produces a reward signal according to the reward function $\mathcal{R}$ that is communicative of the agent's progress through the overall task.

Unlike, the traditional reinforcement learning paradigm, the decision-making problem presented in IL lacks a concrete reward function; in lieu of $\mathcal{R}$, a learner is provided with a dataset of expert demonstrations $\mathcal{D} = \{\tau_1, \tau_2, \dots \tau_N\}$ where each $\tau_i = (s_{i1}, a_{i1}, s_{i2}, a_{i2}, \dots)$ represents the sequence of states and corresponding actions taken by an expert policy, $\pi^*$. Naturally, the goal of an IL algorithm is to synthesize a policy $\pi$ using $\mathcal{D}$, along with access to the MDP $\mathcal{M}$, whose behavior matches that of $\pi^*$.

While the previous section outlines several possible avenues for using $\mathcal{D}$ to arrive at a satisfactory imitation policy, our work focuses on adversarial methods that build around GAIL (Ho & Ermon, 2016). Following from the widespread success of GANs (Goodfellow et al., 2014), GAIL offers a highly-performant approach to IL wherein, at each iteration of the algorithm, transitions sampled from the current imitation policy are first used to update a discriminator, $D_\omega(s, a)$, that acts a binary classifier to distinguish between state-action pairs sampled according to the distributions induced by the expert and student. Subsequently, treating the imitation policy as a generator, policy-gradient reinforcement learning is used to shift the current policy towards expert behavior, issuing higher rewards for those generated state-action pairs that are regarded as belonging to the expert according to $D_\omega(s, a)$. More formally, this minimax optimization follows as

$$\min_\pi \max_\omega \mathbb{E}_{(s,a) \sim \rho^{\pi^*}}[\log(D_\omega(s, a))] + \mathbb{E}_{(s,a) \sim \rho^\pi}[\log(1 - D_\omega(s, a))] \tag{1}$$

where $\rho^{\pi^*}(s, a)$ and $\rho^\pi(s, a)$ denote the undiscounted stationary distributions over state-action pairs for the expert and imitation policies respectively. Here $D_\omega(s, a) = \sigma(V_\omega(s, a))$ where $V_\omega(s, a)$ represents the unconstrained output of a discriminator neural network with parameters $\omega$ and $\sigma(v) = (1 + e^{-x})^{-1}$ denotes the sigmoid activation function. Since the imitation policy only exerts control over the latter term in the above objective, the per-timestep reward function maximized by reinforcement learning is given as $r(s, a, s') = -\log(1 - D_\omega(s, a))$. In practice, an entropy regularization term is often added to the objective when optimizing the imitation policy so as to avoid premature convergence to a suboptimal solution (Mnih et al., 2016; Ho & Ermon, 2016; Neu et al., 2017).

In order to accommodate various observation-only data sources (Zhou et al., 2018) and remove the burden of requiring expert action labels, the imitation from observation setting adjusts the expert demonstration dataset $\mathcal{D}$ such that each trajectory $\tau_i = (s_{i1}, s_{i2}, \dots)$ consists only of expert observation sequences. Retaining the goal of recovering an imitation policy that closely resembles expert behavior, Torabi et al. (2018b) introduce GAIFO as the natural extension of GAIL for matching the state transition distribution of the expert policy. Note that an objective for matching the stationary distribution over expert state transitions enables the provision of per-timestep feedback while simultaneously avoid the issues of temporal alignment that arise when trying to match trajectories directly. The resulting algorithm iteratively finds a solution to the following minimax optimization:

$$\min_\pi \max_\omega \mathbb{E}_{(s,s') \sim \rho^{\pi^*}}[\log(D_\omega(s, s'))] + \mathbb{E}_{(s,s') \sim \rho^\pi}[\log(1 - D_\omega(s, s'))] \tag{2}$$

where $\rho^{\pi^*}(s, s')$ and $\rho^\pi(s, s')$ now denote the analogous stationary distributions over successive state pairs while $D_\omega(s, s') = \sigma(V_\omega(s, s'))$ represents binary classifier over state pairs. Similar to GAIL, the imitation policy is optimized via policy-gradient reinforcement learning with per-timestep rewards computed according to $r(s, a, s') = -\log(1 - D_\omega(s, s'))$ and using entropy regularization as needed.

## 4 APPROACH

In this section, we begin with an overview of $f$-divergences, their connection to GANs, and their impact on IL through the $f$-VIM framework (Ke et al., 2019) (Section 4.1). We then present an alternative view of the framework that transparently exposes the fundamental choice practitioners must make in order to circumvent practical issues that arise when applying $f$-VIM to high-dimensional tasks (Section 4.2). We conclude by presenting our approach for ILfO as $f$-divergence minimization

| Name | Output Activation $g_f$ | $\mathrm{dom}_{f^*}$ | Conjugate $f^*(t)$ | $\mathrm{dom}_{f^{*-1}}$ | Conjugate Inverse $f^{*-1}(t)$ |
|---|---|---|---|---|---|
| Total Variation (TV) | $\frac{1}{2}\tanh(v)$ | $-\frac{1}{2} \leq t \leq \frac{1}{2}$ | $t$ | $-\frac{1}{2} \leq t \leq \frac{1}{2}$ | $t$ |
| Kullback-Leibler (KL) | $v$ | $\mathbb{R}$ | $\exp(t-1)$ | $\mathbb{R}_+$ | $1 + \log(t)$ |
| Reverse KL (RKL) | $-\exp(v)$ | $\mathbb{R}_-$ | $-1 - \log(-t)$ | $\mathbb{R}$ | $-\exp(-1-t)$ |
| GAN | $-\log(1+\exp(-v))$ | $\mathbb{R}_-$ | $-\log(1-\exp(t))$ | $\mathbb{R}_+$ | $\log(1-\exp(-t))$ |

Table 1: Table of various $f$-divergences studied in this work as well as the specific choices of activation function $g_f$ given by Nowozin et al. (2016) and utilized in Ke et al. (2019). Also shown are the convex conjugates, inverse convex conjugates, and their respective domains.

(Section 4.3) followed by a brief discussion of a regularization technique used to stabilize discriminator training in our experiments (Section 4.4).

## 4.1 $f$-DIVERGENCES AND IMITATION LEARNING

The GAIL and GAIFO approaches engage in an adversarial game where the discriminator estimates the divergence between state-action or state transition distributions according to the Jensen-Shannon divergence (Goodfellow et al., 2014). In this work, our focus is on a more general class of divergences, that includes the Jensen-Shannon divergence, known as Ali-Silvey distances or $f$-divergences (Ali & Silvey, 1966; Csiszár et al., 2004). For two distributions $P$ and $Q$ with support over a domain $\mathcal{X}$ and corresponding continuous densities $p$ and $q$, we have the $f$-divergence between them according to:

$$D_f(P||Q) = \int_{\mathcal{X}} q(x)f(\frac{p(x)}{q(x)})dx \tag{3}$$

where $f : \mathbb{R}_+ \to \mathbb{R}$ is a convex, lower-semicontinuous function such that $f(1) = 0$. As illustrated in Table 1, different choices of function $f$ yield well-known divergences between probability distributions. In order to accommodate the tractable estimation of $f$-divergences when only provided samples from $P$ and $Q$, Nguyen et al. (2010) offer an approach for variational estimation of $f$-divergences. Central to their procedure is the use of the convex conjugate function or Fenchel conjugate (Hiriart-Urruty & Lemaréchal, 2004), $f^*$, which exists for all convex, lower-semicontinuous functions $f$ and is defined as the following supremum:

$$f^*(t) = \sup_{u \in \mathrm{dom}_f} \{ut - f(u)\} \tag{4}$$

Using the duality of the convex conjugate ($f^{**} = f$), Nguyen et al. (2010) represent $f(u) = \sup_{t \in \mathrm{dom}_{f^*}} \{tu - f^*(t)\}$ enabling a variational bound:

$$
\begin{aligned}
D_f(P||Q) &= \int_{\mathcal{X}} q(x) \sup_{t \in \mathrm{dom}_{f^*}} \left\{ t\frac{p(x)}{q(x)} - f^*(t) \right\} dx \\
&\geq \sup_{T \in \mathcal{T}} \left( \int_{\mathcal{X}} p(x)T(x)dx - \int_{\mathcal{X}} q(x)f^*(T(x))dx \right) \\
&= \sup_{T \in \mathcal{T}} \left( \mathbb{E}_{x \sim P}[T(x)] - \mathbb{E}_{x \sim Q}[f^*(T(x))] \right)
\end{aligned}
\tag{5}
$$

where $\mathcal{T}$ is an arbitrary class of functions $T : \mathcal{X} \to \mathrm{dom}_{f^*}$. Nowozin et al. (2016) extend the use of this variational lower bound for GANs that utilize arbitrary $f$-divergences, or $f$-GANs. Specifically, the two distributions of interest are the real data distribution $P$ and a synthetic distribution represented by a generative model $Q_\theta$ with parameters $\theta$. The variational function is also parameterized as $T_\omega$ acting as the discriminator. This gives rise to the VDM principle which defines the $f$-GAN objective

$$\min_\theta \max_\omega \mathbb{E}_{x \sim P}[T_\omega(x)] - \mathbb{E}_{x \sim Q_\theta}[f^*(T_\omega(x))] \tag{6}$$

Nowozin et al. (2016) represent the variational function as $T_\omega(x) = g_f(V_\omega(x))$ such that $V_\omega(x) : \mathcal{X} \to \mathbb{R}$ represents the unconstrained discriminator network while $g_f : \mathbb{R} \to \mathrm{dom}_{f^*}$ is an activation

function chosen in accordance with the $f$-divergence being optimized. Table 1 includes the "somewhat arbitrary" but effective choices for $g_f$ suggested by Nowozin et al. (2016) and we refer readers to their excellent work for more details and properties of $f$-divergences and $f$-GANs.

Recently, Ke et al. (2019) have formalized the generalization from GAN to $f$-GAN for the traditional IL problem. They offer the $f$-Variational Imitation ($f$-VIM) framework for the specific case of estimating and then minimizing the divergence between state-action distributions induced by expert and imitation policies:

$$\min_\theta \max_\omega \mathbb{E}_{(s,a)\sim\rho^{\pi^*}} [g_f(V_\omega(s,a))] - \mathbb{E}_{(s,a)\sim\rho^{\pi_\theta}} [f^*(g_f(V_\omega(s,a)))] \tag{7}$$

where $V_\omega : \mathcal{S} \times \mathcal{A} \to \mathbb{R}$ denotes the discriminator network that will supply per-timestep rewards during the outer policy optimization which itself is carried out over policy parameters $\theta$ via policy-gradient reinforcement learning (Sutton et al., 2000). In particular, the per-timestep rewards provided to the agent are given according to $r(s,a,s') = f^*(g_f(V_\omega(s,a)))$.

While Ke et al. (2019) do an excellent job of motivating the use of $f$-divergences for IL (by formalizing the relationship between divergences over trajectory distributions vs. state-action distributions) and connecting $f$-VIM to existing imitation-learning algorithms, their experiments focus on smaller problems to study the mode-seeking/mode-covering aspects of different $f$-divergences and the implications of such behavior depending on the multimodality of the expert trajectory distribution. Meanwhile, in the course of attempting to apply $f$-VIM to large-scale imitation problems, we empirically observe numerical instabilities stemming from function approximation, demanding a reformulation of the framework.

## 4.2 Reparameterizing $f$-VIM

In their presentation of the $f$-VIM framework, Ke et al. (2019) retain the choices for activation function $g_f$ introduced by Nowozin et al. (2016) for $f$-GANs. Recall that these choices of $g_f$ play a critical role in defining the reward function optimized by the imitation policy on each iteration of $f$-VIM, $r(s,a,s') = f^*(g_f(V_\omega(s,a)))$. It is well known in the reinforcement-learning literature that the nature of the rewards provided to an agent have strong implications on learning success and efficiency (Ng et al., 1999; Singh et al., 2010). While the activation choices made for $f$-GANs are suitable given that both optimization problems are carried out by backpropagation, we assert that special care must be taken when specifying these activations (and implicitly, the reward function) for imitation-learning algorithms. A combination of convex conjugate and activation function could induce a reward function that engenders numerical instability or a simply challenging reward landscape, depending on the underlying policy-gradient algorithm utilized (Henderson et al., 2018). Empirically, we found that the particular activation choices for the KL and reverse KL divergences shown in Table 1 (linear and exponential, respectively) produced imitation-learning algorithms that, in all of our evaluation environments, failed to complete execution due to numerical instabilities caused by exploding policy gradients. In the case of the Total Variation distance, the corresponding $f$-GAN activation for the variational function is a $\texttt{tanh}$, requiring a learning agent to traverse a reward interval of $[-1, 1]$ by crossing an intermediate region with reward signals centered around $0$.

To refactor the $f$-VIM framework so that it more clearly exposes the choice of reward function to practitioners and shifts the issues of reward scale away from the imitation policy, we propose uniformly applying an activation function $g_f(v) = f^{*-1}(r(v))$ where $f^{*-1}(t)$ denotes the inverse of the convex conjugate (see Table 1). Here $r$ is effectively a free parameter that can be set according to one of the many heuristics used throughout the field of deep reinforcement learning for maintaining a reasonable reward scale (Mnih et al., 2015; 2016; Henderson et al., 2018) so long as it obeys the domain of the inverse conjugate $\text{dom}_{f^{*-1}}$. In selecting $g_f$ accordingly, the reparameterized saddle-point optimization for $f$-VIM becomes

$$\min_\theta \max_\omega \mathbb{E}_{(s,a)\sim\rho^{\pi^*}} [f^{*-1}(r(V_\omega(s,a)))] - \mathbb{E}_{(s,a)\sim\rho^{\pi_\theta}} [r(V_\omega(s,a))] \tag{8}$$

where the per-timestep rewards used during policy optimization are given by $r(s,a,s') = r(V_\omega(s,a))$. In applying this choice, we shift the undesirable scale of the latter term in VDM towards the discriminator, expecting it to be indifferent since training is done by backpropagation. As one potential instantiation, we consider $r(u) = \sigma(u)$ where $\sigma(\cdot)$ denotes the sigmoid function leading to bounded rewards in the interval $[0, 1]$ that conveniently adhere to $\text{dom}_{f^{*-1}}$ for almost all of

the $f$-divergences examined in this work[1]. In Section 5, we evaluate imitation-learning algorithms with this choice against those using $f$-VIM with the original $f$-GAN activations; we find that, without regard for the scale of rewards and the underlying reinforcement-learning problem being solved, the $f$-GAN activation choices either produce degenerate solutions or completely fail to produce an imitation policy altogether.

### 4.3  $f$-DIVERGENCES AND IMITATION FROM OBSERVATION

Applying the variational lower bound of Nguyen et al. (2010) and the corresponding $f$-GAN extension, we can now present our Variational Imitation from Observation ($f$-VIMO) extension for a general family of ILfO algorithms that leverage the VDM principle in the underlying saddle-point optimization. Since optimization of the generator will continue to be carried out by policy-gradient reinforcement learning, we adhere to our reparameterization of the $f$-VIM framework and present the $f$-VIMO objective as:

$$\min_\theta \max_\omega \mathbb{E}_{(s,s')\sim\rho^{\pi^*}}[f^{*-1}(r(V_\omega(s,s')))] - \mathbb{E}_{(s,s')\sim\rho^{\pi_\theta}}[r(V_\omega(s,s'))] \tag{9}$$

with the per-timestep rewards given according to $r(s,a,s') = r(V_\omega(s,s'))$. We present the full approach as Algorithm 1. Just as in Section 4.2, we again call attention to Line 5 where the discriminator outputs (acting as individual reward signals) scale the policy gradient, unlike the more conventional discriminator optimization of Line 4 by backpropagation; this key difference is the primary motivator for our specific reparameterization of the $f$-VIM framework. Just as in the previous section, we take $r(u) = \sigma(u)$ as a particularly convenient choice of activation given its agreement to the inverse conjugate domains $\text{dom}_{f^{*-1}}$ for many choices of $f$-divergence and we employ this instantiation throughout all of our experiments. We leave the examination of alternative choices for $r$ to future work.

---

**Algorithm 1** $f$-VIMO

1: INPUT: Dataset of expert trajectories $\mathcal{D}$, initial policy and discriminator parameters $\theta_0$ and $\omega_0$, number of iterations $N$, discount factor $\gamma$
2: **for** $i = 0, 1, \ldots, N$ **do**
3:     Sample trajectories from current imitation policy $\tau_i \sim \pi_{\theta_i}$
4:     $\omega_{i+1} = \omega_i + \nabla_\omega\Big(\mathbb{E}_{(s,s')\sim\mathcal{D}}[f^{*-1}(r(V_\omega(s,s')))] - \mathbb{E}_{(s,s')\sim\tau_i}[r(V_\omega(s,s'))]\Big)$
5:     Update $\theta_i$ to $\theta_{i+1}$ via a policy-gradient update with rewards given by $r(V_\omega(s,s'))$:

$$\theta_{i+1} = \theta_i + \mathbb{E}_{(s,a,s')\sim\tau_i}\Big[\nabla_\theta \log(\pi_{\theta_i}(a|s))\mathbb{E}_{\tau_i}[\sum_{t=1}^{\infty}\gamma^{t-1}r(V_\omega(s_{t-1},s_t))|s_0 = s, s_1 = s']\Big]$$

6: **end for**

---

### 4.4  DISCRIMINATOR REGULARIZATION

The refactored version of $f$-VIM presented in Section 4.2 is fundamentally addressing instability issues that may occur on the generator side of adversarial training; in our experiments, we also examine the utility of regularizing the discriminator side of the optimization for improved stability. Following from a line of work examining the underlying mathematical properties of GAN optimization (Roth et al., 2017; 2018; Mescheder et al., 2018), we opt for the simple gradient-based regularization of Mescheder et al. (2018) which (for $f$-VIMO) augments the discriminator loss with the following regularization term:

$$R(\omega) = \frac{\psi}{2}\mathbb{E}_{(s,s')\sim\rho^{\pi^*}}[||\nabla_\omega f^{*-1}(r(V_\omega(s,s')))||^2] \tag{10}$$

where $\psi$ is a hyperparameter controlling the strength of the regularization. The form of this specific penalty follows from the analysis of Roth et al. (2017); intuitively, its purpose is to disincentivize the

---

[1]For Total Variation distance, we use $r(u) = \frac{1}{2}\sigma(u)$ to remain within $\text{dom}_{f^{*-1}}$.

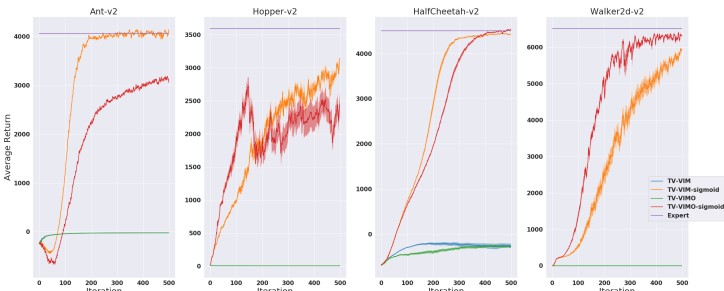

Figure 2: Comparing our TV-VIM and TV-VIMO frameworks (Equations 8 & 9) with sigmoid rewards against the original TV-VIM formulation (Equation 7) and its TV-VIMO counterpart with activations prescribed by Nowozin et al. (2016) (Table 1).

discriminator from producing a non-zero gradient that shifts away from the Nash equilibrium of the minimax optimization when presented with a generator that perfectly matches the true data distribution. While originally developed for traditional GANs and shown to empirically exhibit stronger convergence properties over Wasserstein GANs (Gulrajani et al., 2017), this effect is still desirable for the adversarial IL setting where the reward function (discriminator) used for optimizing the imitation policy should stop changing once the expert state-transition distribution has been matched. In practice, we compare $f$-VIM and $f$-VIMO both with and without the use of this regularization term and find that $R(\omega)$ can improve the stability and convergence of $f$-VIMO across almost all domains.

## 5 EXPERIMENTS

We examine four instantiations of the $f$-VIM and $f$-VIMO frameworks (as presented in Sections 4.2 and 4.3) corresponding to imitation algorithms with the following choices of $f$-divergence: GAN, Kullback-Leibler, reverse KL, and Total Variation. We conduct our evaluation across four MuJoCo environments (Todorov et al., 2012) of varying difficulty: Ant, Hopper, HalfCheetah, and Walker (see the Appendix for more details on individual environments).

The core questions we seek to answer through our empirical results are as follows:

1. What are the implications of the choice of activation for the variational function in $f$-VIM on imitation policy performance?

2. Do $f$-divergences act as a meaningful axis of variation for IL and ILfO algorithms?

3. What is the impact of discriminator regularization on the stability and convergence properties of $f$-VIM/$f$-VIMO?

4. How does the impact of different $f$-divergences vary with the amount of expert demonstration data provided?

To answer the first three questions above, we report the average total reward achieved by the imitation policy throughout the course of learning with rewards as defined by the corresponding OpenAI Gym environment (Brockman et al., 2016). Shading in all plots denote $95\%$ confidence intervals computed over 10 random trials with 10 random seeds. Expert demonstration datasets of 50 trajectories were collected from agents trained via Proximal Policy Optimization (PPO) (Schulman et al., 2017); 20 expert demonstrations were randomly subsampled at the start of learning and held fixed for the duration of the algorithm. We also utilize PPO as the underlying reinforcement-learning algorithm for training the imitation policy with a clipping parameter of $0.2$, advantage normalization, entropy regularization coefficient $1e^{-3}$, and the Adam optimizer (Kingma & Ba, 2014). Just as in Ho & Ermon (2016) we use a discount factor of $\gamma = 0.995$ and apply Generalized Advantage Estimation (Schulman et al., 2015) with parameter $\lambda = 0.97$. We run both $f$-VIM and $f$-VIMO for a total of $500$ iterations, collecting $50000$ environment samples per iteration. The policy and discriminator architectures are identically two separate multi-layer perceptrons each with two hidden layers of 100 units separated by $\tanh$ nonlinearities. A grid search was used for determining the initial

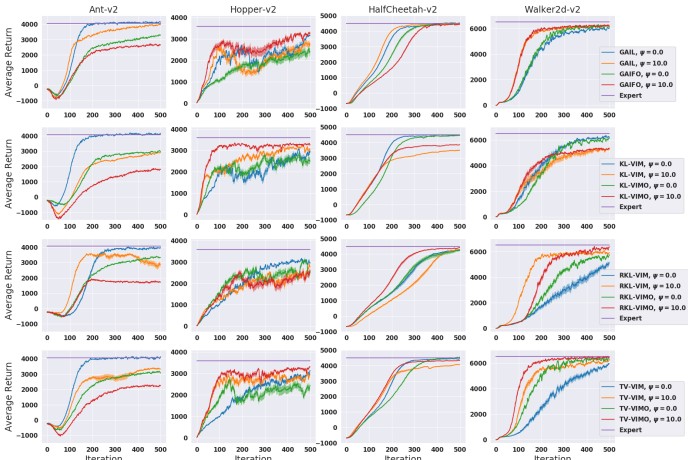

Figure 3: Comparing $f$-VIM and $f$-VIMO across four MuJoCo environments with $f$-divergences: GAN, Kullback-Leibler (KL), reverse KL (RKL), and Total Variation distance (TV). We also examine the effect of discriminator regularization (Equation 10) with $\psi = 10$ per Mescheder et al. (2018).

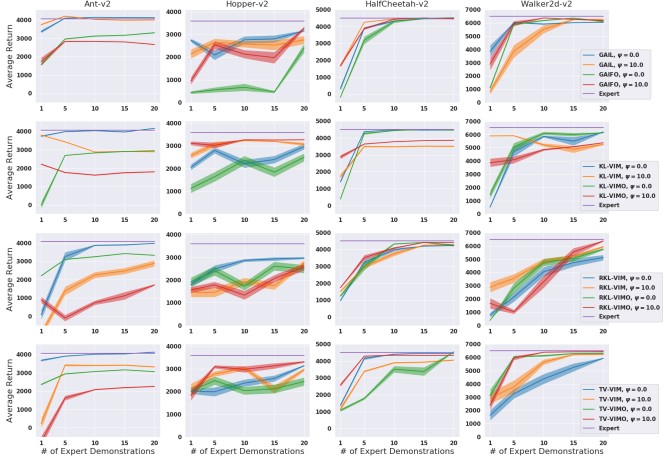

Figure 4: Evaluating $f$-VIM and $f$-VIMO with and without regularization across four MuJoCo environments with varying amounts of expert demonstration data.

learning rate, number of PPO epochs, and number of epochs used for discriminator training (please see the Appendix for more details) and we report results for the best hyperparameter settings.

To address our final question, we take the best hyperparameter settings recovered when given 20 expert demonstrations and re-run all algorithms with $\{1, 5, 10, 15\}$ expert demonstrations that are randomly sampled at the start of each random trial and held fixed for the duration of the algorithm. We then record the average return of the final imitation policy for each level of expert demonstration.

## 6 RESULTS & DISCUSSION

To highlight the importance of carefully selecting the variational function activation $g_f$ and validate our modifications to the $f$-VIM framework, we present results in Figure 2 comparing to the original $f$-VIM framework of Ke et al. (2019) and its natural ILfO counterpart. Activation functions for the original methods are chosen according to the choices outlined in Ke et al. (2019); Nowozin et al. (2016). In our experiments using the KL and reverse KL divergences, we found that none of the trials reached completion due to numerical instabilities caused by exploding policy gradients. Consequently, we only present results for the Total Variation distance. We observe that under the original $f$-GAN activation selection, we fail to produce meaningful imitation policies with learning stagnating after 100 iterations or less. As previously mentioned, we suspect that this stems from the use of $\texttt{tanh}$ with TV leading to a dissipating reward signal.

We present results in Figure 3 to assess the utility of varying the choice of divergence in $f$-VIM and $f$-VIMO across each domain. In considering the impact of $f$-divergence choice, we find that most of the domains must be examined in isolation to observe a particular subset of $f$-divergences that stand out. In the IL setting, we find that varying the choice of $f$-divergence can yield different learning curves but, ultimately, produce near-optimal (if not optimal) imitation policies across all domains. In contrast, we find meaningful choices of $f$-divergence in the ILfO setting including {KL, TV} for Hopper, RKL for HalfCheetah, and {GAN, TV} for Walker. We note that the use of discriminator regularization per Mescheder et al. (2018) is crucial to achieving these performance gains, whereas the regularization generally fails to help performance in the IL setting. This finding is supportive of the logical intuition that ILfO poses a fundamentally more-challenging problem than standard IL.

As a negative result, we find that the Ant domain (the most difficult environment with $\mathcal{S} \subset \mathbb{R}^{111}$ and $\mathcal{A} \subset \mathbb{R}^8$) still poses a challenge for ILfO algorithms across the board. More specifically, we observe that discriminator regularization hurts learning in both the IL and ILfO settings. While the choice of RKL does manage to produce a marginal improvement over GAIFO, the gap between existing state-of-the-art and expert performance remains unchanged. It is an open challenge for future work to either identify the techniques needed to achieve optimal imitation policies from observations only or characterize a fundamental performance gap when faced with sufficiently large observation spaces.

In Figure 4, we vary the total number of expert demonstrations available during learning and observe that certain choices of $f$-divergences can be more robust in the face of less expert data, both in the IL and ILfO settings. We find that KL-VIM and TV-VIM are slightly more performant than GAIL when only provided with a single expert demonstration. Notably, in each domain we see that certain choices of divergence for $f$-VIMO do a better job of residing close to their $f$-VIM counterparts suggesting that future improvements may come from examining $f$-divergences in the small-data regime. This idea is further exemplified when accounting for results collected while using discriminator regularization (Mescheder et al., 2018). We refer readers to the Appendix for the associated learning curves.

Our work leaves many open directions for future work to close the performance gap between student and expert policies in the ILfO setting. While we found the sigmoid function to be a suitable instantiation of our framework, exploring alternative choices of variational function activations could prove useful in synthesizing performant ILfO algorithms. Alternative choices of $f$-divergences could lead to more substantial improvements than the choices we examine in this paper. Moreover, while this work has a direct focus on $f$-divergences, Integral Probability Metrics (IPMs) (Müller, 1997; Gretton et al., 2012) represent a distinct but well-established family of divergences between probability distributions. The success of Total Variation distance in our experiments, which doubles as both a $f$-divergence and IPM (Sriperumbudur et al., 2009), is suggestive of future work building IPM-based ILfO algorithms (Sun et al., 2019).

## 7 CONCLUSION

In this work, we present a general framework for imitation learning and imitation learning from observations under arbitrary choices of $f$-divergence. We empirically validate a single instantiation of our framework across multiple $f$-divergences, demonstrating that we overcome the shortcomings of prior work and offer a wide class of IL and ILfO algorithms capable of scaling to larger problems.

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

# A    RELATED WORK

## A.1    LEARNING FROM DEMONSTRATION

Our work broadly falls within the category of Learning from Demonstration (LfD) (Schaal, 1997; Atkeson & Schaal, 1997; Argall et al., 2009), where an agent must leverage demonstration data (typically provided as trajectories, each consisting of expert state-action pairs) to produce an imitation policy that correctly captures the demonstrated behavior. Within the context of LfD, a finer distinction can be made between behavioral cloning (BC) (Bain & Sommut, 1999; Pomerleau, 1989) and inverse reinforcement learning (IRL) (Ng et al.; Abbeel & Ng, 2004; Syed & Schapire, 2007; Ziebart et al., 2008; Finn et al., 2016; Ho & Ermon, 2016) approaches; BC approaches view the demonstration data as a standard dataset of input-output pairs and apply traditional supervised-learning techniques to recover an imitation policy. Alternatively, IRL-based methods synthesize an estimate of the reward function used to train the expert policy before subsequently applying a reinforcement-learning algorithm (Sutton & Barto, 1998; Abbeel & Ng, 2004) to recover the corresponding imitation policy. Although not a focus of this work, we also acknowledge the myriad of approaches that operate at the intersection of IL and reinforcement learning or augment reinforcement learning with IL (Rajeswaran et al., 2017; Hester et al., 2018; Salimans & Chen, 2018; Sun et al., 2018; Borsa et al., 2019; Tirumala et al., 2019).

While BC approaches have been successful in some settings (Niekum et al., 2015; Giusti et al., 2016; Bojarski et al., 2016), they are also susceptible to failures stemming from covariate shift where minute errors in the actions of the imitation policy compound and force the agent into regions of the state space not captured in the original demonstration data. While some preventative measures for covariate shift do exist (Laskey et al., 2017b), a more principled solution can be found in methods like DAgger (Ross et al., 2011) and its descendants (Ross & Bagnell, 2014; Sun et al., 2017; Le et al., 2018) that remedy covariate shift by querying an expert to provide on-policy action labels. It is worth noting, however, that these approaches are only feasible in settings that admit such online interaction with an expert (Laskey et al., 2016) and, even then, failure modes leading to poor imitation policies do exist (Laskey et al., 2017a).

The algorithms presented in this work fall in with IRL-based approaches to IL. Early successes in this regime tend to rely on hand-engineered feature representations for success (Abbeel & Ng, 2004; Ziebart et al., 2008; Levine et al., 2011). Only in recent years, with the aid of deep neural networks, has there been a surge in the number of approaches that are capable of scaling to the raw, high-dimensional observations found in real-world control problems (Finn et al., 2016; Ho & Ermon, 2016; Duan et al., 2017; Li et al., 2017; Fu et al., 2017; Kim & Park, 2018). Our work focuses attention exclusively on adversarial methods for their widespread effectiveness across a range of imitation tasks without requiring interactive experts (Ho & Ermon, 2016; Li et al., 2017; Fu et al., 2017; Kostrikov et al., 2018); at the heart of these methods is the Generative Adversarial Imitation Learning (GAIL) (Ho & Ermon, 2016) approach which produces high-fidelity imitation policies and achieves state-of-the-art results across numerous continuous-control benchmarks by leveraging the

expressive power of Generative Adversarial Networks (GANs) (Goodfellow et al., 2014) for modeling complex distributions over a high-dimensional support. From an IRL perspective, GAIL can be viewed as iteratively optimizing a parameterized reward function (discriminator) that, when used to optimize an imitation policy (generator) via policy-gradient reinforcement learning (Sutton et al., 2000), allows the agent to shift its own behavior closer to that of the expert. From the perspective of GANs, this is achieved by discriminating between the respective distributions over state-action pairs visited by the imitation and expert policies before training a generator to fool the discriminator and induce a state-action visitation distribution similar to that of the expert.

While a large body of prior work exists for IL, numerous recent works have drawn attention to the more challenging problem of imitation learning from observation (Sermanet et al., 2017; Liu et al., 2018; Goo & Niekum, 2018; Kimura et al., 2018; Torabi et al., 2018a;b; Edwards et al., 2019; Sun et al., 2019). In an effort to more closely resemble observational learning in humans and leverage the wealth of publicly-available, observation-only data sources, the ILfO problem considers learning from expert demonstration data where no expert action labels are provided. Many early approaches to ILfO use expert observation sequences to learn a semantic embedding space so that distances between observation sequences of the imitation and expert policies can serve as a cost signal to be minimized via reinforcement learning (Gupta et al., 2017; Sermanet et al., 2017; Dwibedi et al., 2018; Liu et al., 2018). In contrast, Torabi et al. (2018a) introduce Behavioral Cloning from Observation (BCO) which leverages state-action trajectories collected under a random policy to train an inverse dynamics model for inferring the action responsible for a transition between two input states (assuming the two represent a state and next-state pair). With this inverse model in hand, the observation-only demonstration data can be converted into the more traditional dataset of state-action pairs over which standard BC can be applied. Recognizing the previously discussed limitations of BC approaches, Torabi et al. (2018b) introduce the natural GAIL counterpart for ILfO, Generative Adversarial Imitation from Observation (GAIFO); GAIFO is identical to GAIL except the distributions under consideration in the adversarial game are over state transitions (state and next-state pairs), as opposed to state-action pairs requiring expert action labels. While Torabi et al. (2018b) offer empirical results for continuous-control tasks with low-dimensional features as well as raw image observations, GAIFO falls short of expert performance in both settings leaving an open challenge for scalable ILfO algorithms that achieve expert performance across a wide spectrum of tasks. A central question of this work is to explore how alternative formulations of the GAN objective that underlies GAIFO might yield superior ILfO algorithms. For a more in-depth survey of ILfO approaches, we refer readers to Torabi et al. (2019).

## A.2 GENERATIVE ADVERSARIAL NETWORKS

With a focus on generative-adversarial methods for IL, this work leverages several related ideas in the GAN literature for offering alternative formulations as well as improving understanding of their underlying mathematical foundations (Li et al., 2015; Dziugaite et al., 2015; Zhao et al., 2016; Nowozin et al., 2016; Roth et al., 2017; Arjovsky et al., 2017; Gulrajani et al., 2017; Roth et al., 2018; Mescheder et al., 2018). Critical to the ideas presented in many of these previous works is an understanding that discriminator networks are estimating a divergence between two probability distributions of interest, usually taken to be the real data distribution and the fake or synthetic distribution represented by the generator. Formal characterizations of this divergence, either by Integral Probability Metrics (IPMs) (Müller, 1997; Gretton et al., 2012) or $f$-divergences (Ali & Silvey, 1966; Csiszár et al., 2004; Liese & Vajda, 2006), yield different variations on the classic GAN formulation which is itself a slight variation on the Jensen-Shannon (JS) divergence (Li et al., 2015; Dziugaite et al., 2015; Zhao et al., 2016; Nowozin et al., 2016; Arjovsky et al., 2017; Gulrajani et al., 2017). Following from work by Nowozin et al. (2016) to generalize the GAN objective to arbitrary $f$-divergences, Ke et al. (2019) offer a generalization of GAIL to an arbitrary choice of $f$-divergence for quantifying the gap between the state-action visitation distributions of the imitation and expert policies; moreover, Ke et al. (2019) propose a unifying framework for IL, $f$-Variational IMitation ($f$-VIM), in which they highlight a correspondence between particular choices of $f$-divergences and existing IL algorithms (specifically BC $\Longleftrightarrow$ Kullback-Leibler (KL) divergence, DAgger $\Longleftrightarrow$ Total-Variation distance, and GAIL $\Longleftrightarrow$ JS-divergence [2]). While Ke et al. (2019) focus on providing empirical results in smaller toy problems to better understand the interplay between $f$-divergence

---

[2] The discriminator loss optimized in the original GAN formulation is $2 \cdot D_{JS} - \log(4)$ where $D_{JS}$ denotes the Jensen-Shannon divergence (Goodfellow et al., 2014; Nowozin et al., 2016).

choice and the multimodality of the expert trajectory distribution, we provide an empirical evaluation of their $f$-VIM framework across a range of continous control tasks in the Mujoco domain (Todorov et al., 2012). Empirically, we find that some of the design choices $f$-VIM inherits from the original $f$-GAN work (Nowozin et al., 2016) are problematic when coupled with adversarial IL and training of the generator by policy-gradient reinforcement learning, instead of via direct backpropagation as in traditional GANs. Consequently, we refactor their framework to expose this point and provide one practical instantiation that works well empirically. We then go on to extend the $f$-VIM framework to the IFO problem ($f$-VIMO) and evaluate the resulting algorithms empirically against the state-of-the-art, GAIFO.

## B EXPERIMENT DETAILS

Here we provide details of the MuJoCo environments (Todorov et al., 2012) used in our experiments as well as the details of the hyperparameter search conducted for all algorithms (IL and ILfO) presented.

### B.1 MUJOCO ENVIRONMENTS

All environments have continuous observation and action spaces of varying dimensionality (as shown below). All algorithms evaluated in each environment were trained for a total of 500 iterations, collecting $50,000$ environment transitions per iteration.

| Task | Observation Space | Action Space |
|---|---|---|
| Ant-v2 | $\mathbb{R}^{111}$ | $\mathbb{R}^8$ |
| Hopper-v2 | $\mathbb{R}^{11}$ | $\mathbb{R}^3$ |
| HalfCheetah-v2 | $\mathbb{R}^{17}$ | $\mathbb{R}^6$ |
| Walker2d-v2 | $\mathbb{R}^{17}$ | $\mathbb{R}^6$ |

### B.2 HYPERPARAMETERS

Below we outline the full set of hyperparameters examined for all experiments presented in this work. We conducted a full grid search over 10 random trials with 10 random seeds and report results for the best hyperparameter setting.

| Hyperparameter | Values |
|---|---|
| Discriminator learning rate | $\{1e^{-4}, 1e^{-3}\}$ |
| PPO epochs | $\{5, 10\}$ |
| Discriminator epochs | $\{1, 5, 10\}$ |

Preliminary experiments were conducted to test smaller values for PPO epochs and policy learning rates before settling on the grid shown above.

# C ADDITIONAL RESULTS

## C.1 UNREGULARIZED $f$-VIM/VIMO

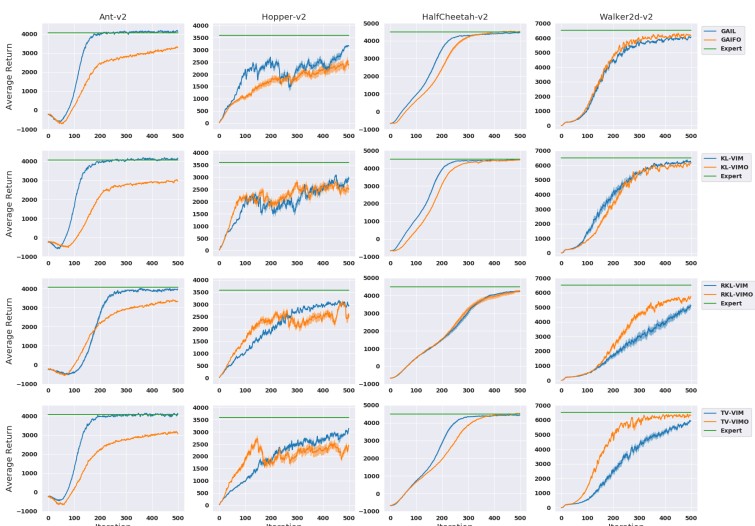

Figure 5: Comparing $f$-VIM and $f$-VIMO across four MuJoCo environments with $f$-divergences: GAN, Kullback-Leibler (KL), reverse KL (RKL), and Total Variation distance (TV).

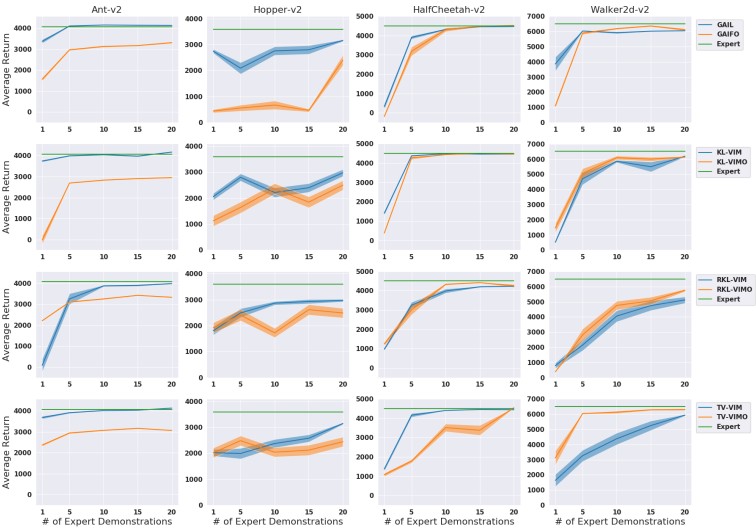

Figure 6: Evaluating $f$-VIM and $f$-VIMO across four MuJoCo environments with varying amounts of expert demonstration data.

## C.2 SAMPLE COMPLEXITY LEARNING CURVES

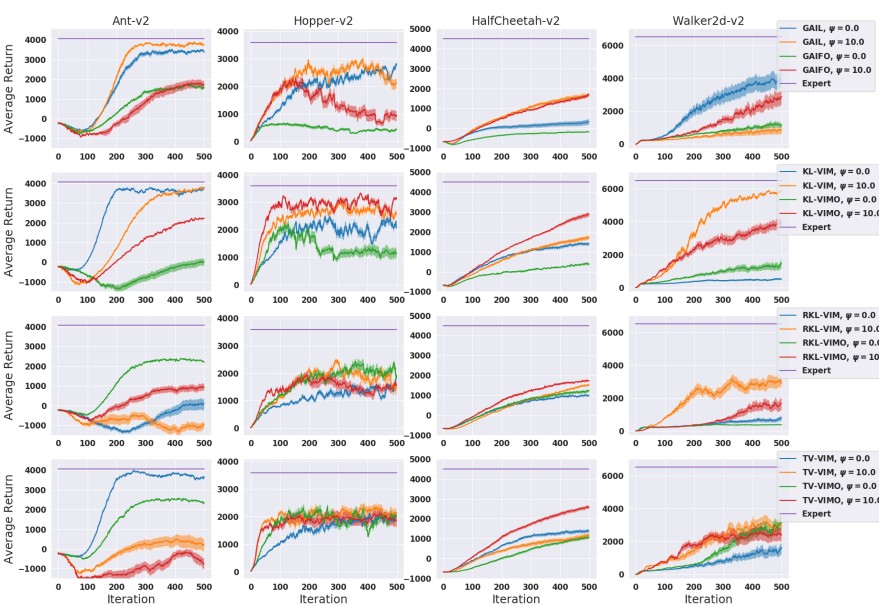

Figure 7: Learning curves for $f$-VIM and $f$-VIMO across four MuJoCo environments using only 1 expert demonstration.

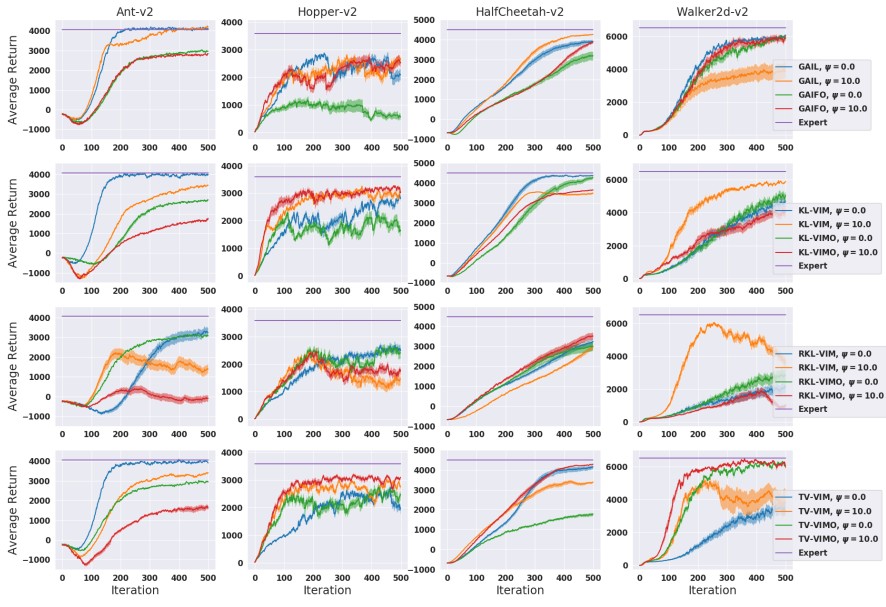

Figure 8: Learning curves for $f$-VIM and $f$-VIMO across four MuJoCo environments using only 5 expert demonstrations.

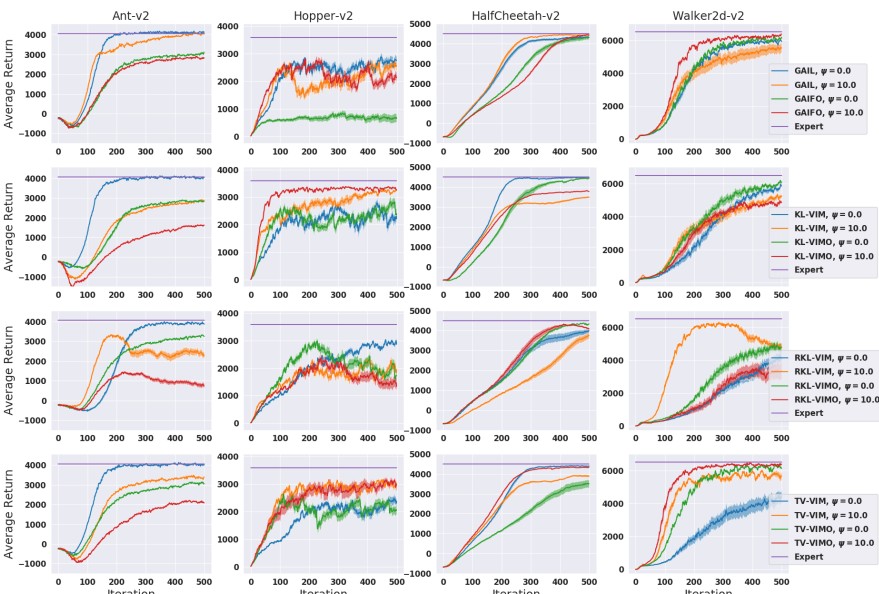

Figure 9: Learning curves for $f$-VIM and $f$-VIMO across four MuJoCo environments using only 10 expert demonstrations.

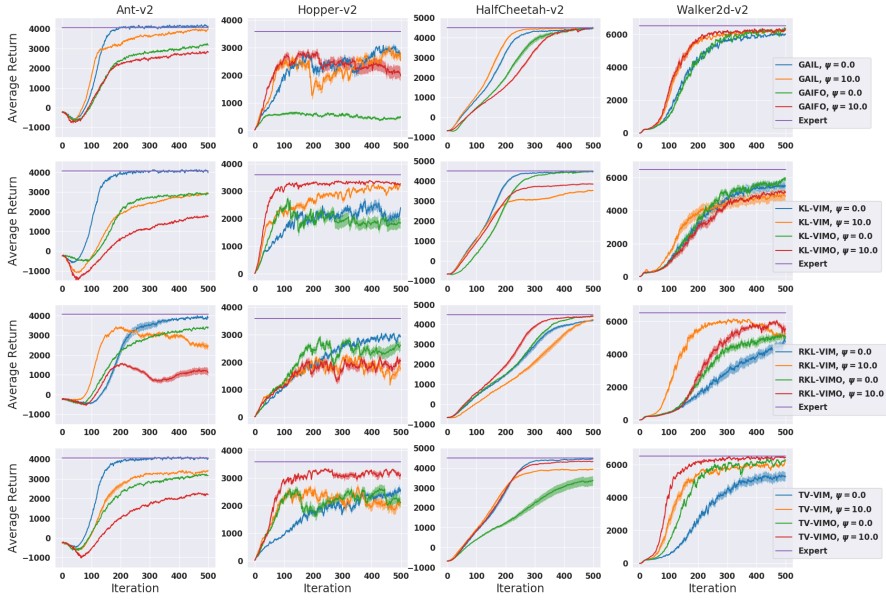

Figure 10: Learning curves for $f$-VIM and $f$-VIMO across four MuJoCo environments using only 15 expert demonstrations.

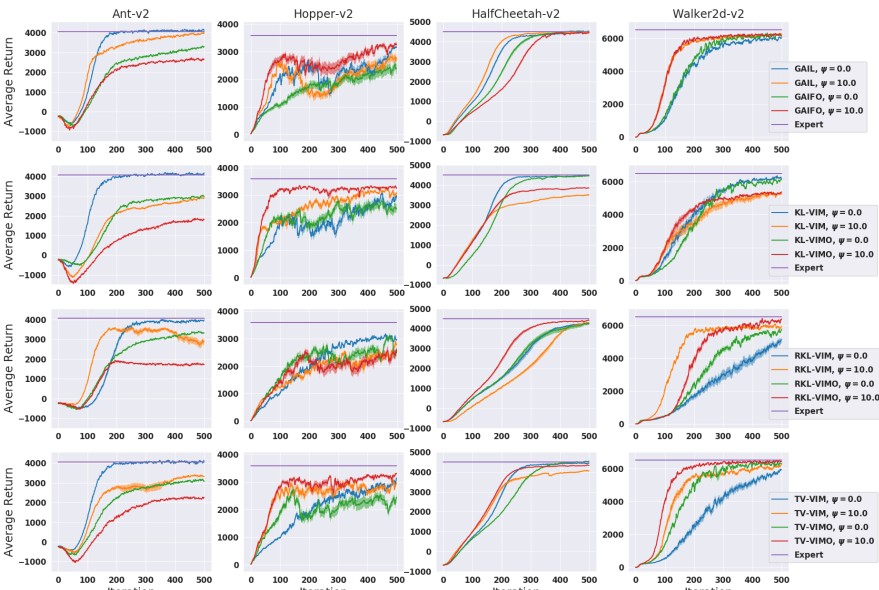

Figure 11: Learning curves for $f$-VIM and $f$-VIMO across four MuJoCo environments using only 20 expert demonstrations.

## C.3   $f$-DIVERGENCE VARIATIONAL BOUND SWAP

Throughout this paper, we advocate for the use of the following variational lower bound to the $f$-divergence for both $f$-VIM and $f$-VIMO:

$$D_f(\rho^{\pi^*}||\rho^{\pi_\theta}) \geq \min_\theta \max_\omega \mathbb{E}_{(s,s')\sim\rho^{\pi^*}}[f^{*-1}(r(V_\omega(s,s')))] - \mathbb{E}_{(s,s')\sim\rho^{\pi_\theta}}[r(V_\omega(s,s'))] \quad (11)$$

In particular, we value the above form as it clearly exposes the choice of reward function for the imitation policy as a free parameter that, in practice, has strong implications for the stability and convergence of adversarial IL/ILfO algorithms. Alternatively, one may consider appealing to the original lower bound of Nguyen et al. (2010), used in $f$-GANs (Nowozin et al., 2016) unmodified, but swapping the positions of the two distributions:

$$D_f(\rho^{\pi_\theta}||\rho^{\pi^*}) \geq \min_\theta \max_\omega \mathbb{E}_{(s,s')\sim\rho^{\pi_\theta}}[g_f(V_\omega(s,s'))] - \mathbb{E}_{(s,s')\sim\rho^{\pi^*}}[f^*(g_f(V_\omega(s,s')))] \quad (12)$$

Consequently, the term in this lower bound pertaining to the imitation policy is now similar to that of the bound in Equation 11; namely, an almost arbitrary activation function, $g_f$, applied to the output of the variational function (discriminator) $V_\omega$. The difference being that the codomain of $g_f$ must obey the domain of the convex conjugate, $f^*$, while the codomain of $r$ must respect the domain of the inverse convex conjugate, $f^{*-1}$.

We evaluate these two choices empirically below for the specific choice of the KL-divergence in the Ant and Hopper domains (the two most difficult domains of our evaluation). We find that the original unswapped bound in Equation 11 used throughout this paper outperforms the variants with the distributions swapper, for both the IL and ILfO settings. Crucially, we find that the KL-VIM in the Ant domain no longer achieves expert performance while optimizing the swapped bound.

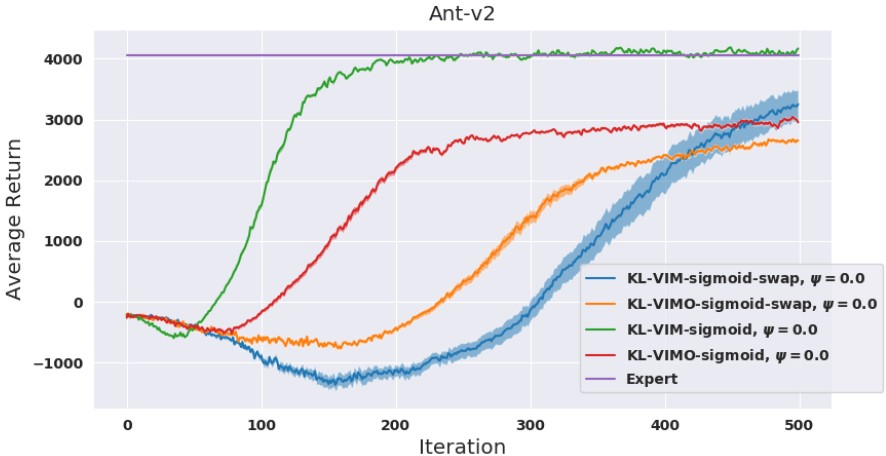

Figure 12: Learning curves for KL-VIM and KL-VIMO in Ant with 20 expert demonstrations using the regular and swapped variational lower bound.

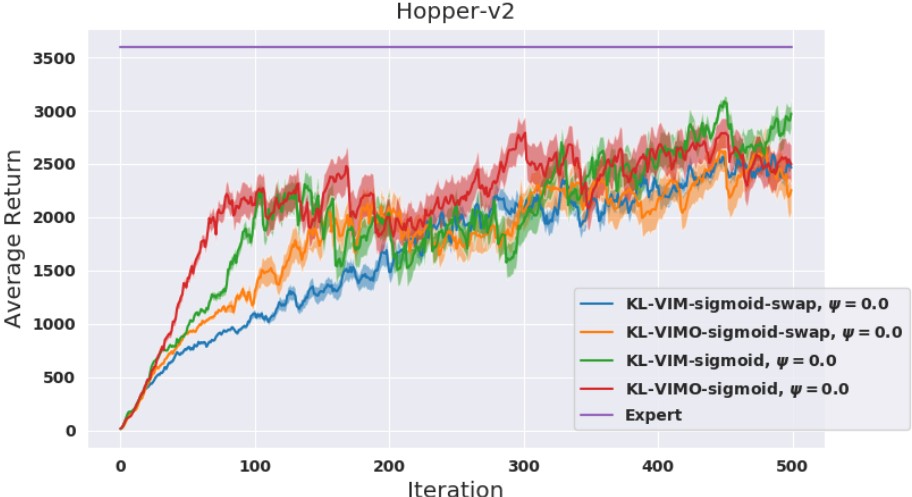

Figure 13: Learning curves for KL-VIM and KL-VIMO in Hopper with 20 expert demonstrations using the regular and swapped variational lower bound.

