# OpenReview forum: "Reparameterized Variational Divergence Minimization for Stable Imitation"
_ICLR.cc/2020/Conference — Reject_

### Official Review · AnonReviewer2 · 2019-10-18
**Official Blind Review #2**

**Rating:** 1

**Review:**

* Summary:
The paper proposes an IL method based on the f-divergence. Specifically, the paper extends f-VIM (Ke et al., 2019), which uses the f-divergence for IL, by using a sigmoid function for discriminator output’s activation function. This choice of activation function yields an alternative objective function, where the reward function for an RL agent does not directly depend on the convex conjugate of the function f; the paper claims that this independency improves stability of policy learning. This proposed method is named f-VIM-sigmoid. The paper extends f-VIM-sigmoid to the setting of IL with observation and proposes f-VIMO-sigmoid. Experiments on Mujoco locomotion tasks show that f-VIM-sigmoid and f-VIMO-sigmoid perform better than existing methods.

* Rating:
The paper proposes a simple but interesting approach to improve stability of adversarial IL. However, the paper has issues regarding baseline methods, motivation, supports of the claim, and experiments (see below). These issues should be addressed. At the present, I vote for rejection.

* Major comments:
- Discussion and comparing against a simple baseline method based on swapping distributions:
To make the reward function be independent of the convex conjugate f*, it is possible to simply swapping the distributions P and Q in the definition of the f-divergence. More specifically, instead of minimizing D_f(P||Q), we can minimize D_f(Q||P), where P is a data distribution and Q is a generator. In this case, pi* and pi_theta in Eq. (7) swap, and the RL agent minimizes the cost function g_f(V_w(s,a)). This cost function does not directly depend on f*, similarly to the reward function r(V_w(s,a)) in Eq. (8). This swapping is simpler and more flexible than re-parameterizing, while achieving the same goal as f-VIM-sigmoid. This swapping method should be discussed and compared against the proposed methods.

- Need stronger baseline methods for ILfO:
The paper should evaluate f-VIMO-sigmoid against stronger baselines, e.g., forward adversarial IL (Sun et al., 2019) which outperforms GAIL-based methods in the ILfO setting.

[1] Wen Sun, Anirudh Vemula, Byron Boots, and J Andrew Bagnell. Provably efficient imitation learning from observation alone. ICML, 2019.

- Using the f-divergence for ILfO is not well motivated:
The paper does not provide good motivations for using f-divergence in ILfO. This makes the paper quite difficult to follow, since there is no connection between f-divergence and ILfO.

- The experiments focus on evaluating existing methods rather than the proposed methods:
Specifically, the proposed methods are evaluated with only one choice of the divergence (TV) in Figure 2. Meanwhile, most of Section 6 and results (Figure 3 and 4, and additional results in the appendix) focus on evaluating the existing methods (f-VIM and f-VIMO) with different choices of divergence.

- The experiments in Figure 2 do not support the claim regarding stability:
The paper claims to improve stability of IL by using the proposed re-parameterization. However, the experimental results do not support this claim, and the questions asked in Section 5 are not related to this claimed. Instead, it seems that re-parameterization helps avoiding local optima (possibly due to a biased reward function, see below), while stability is improved by regularizing the discriminator. I could not see how the re-parameterization improves the policy stability as claimed.

- The experiments in Figure 2 seem unfair, since TV-VIM-sigmoid incorporates priors about survival bonuses:
Specifically, TV-VIM-sigmoid uses sigmoid which yields strictly positive rewards, while TV-VIM uses tanh which yields positive and negative rewards. As discussed by Kostrikov et al., 2019, using strictly positive rewards incorporate strong priors about the survival bonuses, which exist in the locomotion task used in the experiments. Therefore, TV-VIM-sigmoid uses strong priors while TV-VIM does not. In order to make the comparison fairer, I suggest the authors to evaluate TV-VIM with sigmoid reward output, or include environments that do not have survival bonuses.

[2] Ilya Kostrikov, Kumar Krishna Agrawal, Debidatta Dwibedi, Sergey Levine, and Jonathan Tompson. Discriminator-actor-critic: Addressing sample inefficiency and reward bias in adversarial imitation learning. ICLR, 2019

* Minor comments:
- The abstract is long and could be shorten.
- Figures are too small and difficult to see, especially the legends.
- Table 1 should describe the form of f in addition to its conjugate.
- The title of the Algorithm 1 should be f-VIMO-sigmoid instead of f-VIMO.

** Update after response.
I read the response. I thank the authors for clarifying the claims as well as the new experiments with the swap formulation. However, improving clarity of the claims is considered a major revision. I still keep the vote of rejection.

Regarding reward bias. As the authors acknowledge, the improvement achieved by using reparameterization+sigmoid can be explained by two equally-plausible reasons: 1) reparameterization+sigmoid improves stability (as claimed) and 2) sigmoid gives biased rewards. The issue here is that we do not know which is the actual reason, given the current experiments in the paper. As I commented, evaluating TV-VIM with sigmoid but without reparameterization will help address this issue.


**Experience Assessment:**

I have published one or two papers in this area.

**Review Assessment: Checking Correctness Of Derivations And Theory:**

I carefully checked the derivations and theory.

**Review Assessment: Checking Correctness Of Experiments:**

I assessed the sensibility of the experiments.

**Review Assessment: Thoroughness In Paper Reading:**

I read the paper at least twice and used my best judgement in assessing the paper.

---

> ### Author Response · Authors · 2019-11-08
> **Response to Reviewer #2**
>
> “Discussion and comparing against a simple baseline method based on swapping distributions” — while it is tempting to simply swap the distributions over which expectations are taken in Equation 7, this choice does have nontrivial implications for the overall f-divergence objective being minimized. In particular, we feel that this choice could be viable but perhaps ineffective due to the mode-seeking/mode-covering differences of swapping the positions of the distributions. We are currently running experiments so that we can better report on this.
>
> “Need stronger baseline methods for ILfO: “ — we note that further ILfO baselines beyond GAIFO would only dilute the signal of interest for this work in establishing the effect of f-divergence choice in the context of ILfO problems. The reviewer suggests a comparison to FAIL (Sun et al., 2019) which assumes a time-factored observation space (an assumption not made in this paper) and a non-stationary policy. In practice, this results in the need for solving solving H distribution-matching (GAN) problems (where H is the finite horizon of the MDP) to recover H per-timestep policies. While the provable guarantees of FAIL are admirable, the computational expense of such a baseline is obviously impractical, something confirmed by the chosen empirical evaluation of the FAIL paper itself which further assumes a reproducing kernel Hilbert space (RKHS) to allow for closed-form computation of all divergences under the maximum mean discrepancy; the potential use of such integral probability metrics, while interesting, is not a focus of this work.
>
> “Using the f-divergence for ILfO is not well motivated: “ — as mentioned in our response to all reviewers, the core hypothesis of this work is that alternative f-divergences may yield more performant ILfO algorithms, akin to the qualitative improvements of f-GANs (Nowozin et al., 2016) over traditional GANs. Again, this hypothesis mirrors past ideas in the literature while being unique to our setting and proving to be true empirically for a subset of the evaluation tasks. We would also refer the reviewers to the f-MAX paper (recent best paper at CoRL 2019) which proposes a similar framework to Ke et al. (2019). A direct quote from the abstract: “In this work, we present a unified probabilistic perspective on IL algorithms based on divergence minimization. We present f-MAX, an f-divergence generalization of AIRL [1], a state-of-the-art IRL method. f-MAX enables us to relate prior IRL methods such as GAIL [2] and AIRL [1], and understand their algorithmic properties. Through the lens of divergence minimization we tease apart the differences between BC and successful IRL approaches, and empirically evaluate these nuances on simulated high-dimensional continuous control domains.”  It is precisely because we can see these popular IL algorithms as specific members of a broader family of algorithms that such insights can be derived; beginning this path for a problem that is as challenging as ILfO is crucial for driving progress.
>
> “The experiments focus on evaluating existing methods rather than the proposed methods:” — as mentioned in the response to all reviewers, we cannot stress enough that the only existing ILfO method evaluated in this work is GAIFO; all others are novel contributions of this work.
>
> “The experiments in Figure 2 do not support the claim regarding stability:” — Figure 2 offers a clear, concrete example highlighting how our reparameterization improves imitation policy stability, ultimately leading to a performant policy for the case of total variation distance. As for the KL and RKL divergences, we cannot report results for divergences that failed to reach completion due to numerical instabilities; preliminary experiments to employ gradient norm clipping with threshold values common to deep reinforcement learning were found to be completely ineffective. Note that a naive, brute-force solution to this problem does exist in the form of grid searching over all possible thresholds for gradient norm clipping of the policy in order to find one that is as large as possible without succumbing to numerical errors. Obviously, the computational infeasibility of such a search across multiple choices of f-divergence is less than desirable and our reparameterization offers a stable optimization solution without relying on such computational inefficiency, which does constitute an important contribution.

---

> > ### Author Response · Authors · 2019-11-08
> > **Response to Reviewer #2 (continued)**
> >
> >
> > “The experiments in Figure 2 seem unfair, since TV-VIM-sigmoid incorporates priors about survival bonuses: “ —  We agree with the reviewer that the sigmoid reward used in Figure 2 does satisfy the second example of bias reward functions listed by Kostrikov et al. (2019). However, the reviewer is only conjecturing that this bias accounts for the gap between TV-VIM and TV-VIM-sigmoid; in the paper, we posit a different, equally-plausible explanation. Namely, that tanh naturally requires an imitation policy to gradually move from a region of negative reward (the lower range of tanh) to a region of positive reward by crossing an intermediate region of 0 reward. Given the nature of adversarial imitation learning, we know that this progression is monotonic (that is, we know imitation policies will start poorly in the negative region and gradually improve, moving towards the positive region). We assert that this disappearing reward signal is what causes learning in TV-VIM to stagnate. Notice that solution employed by Kostrikov et al. (2019) to resolve reward bias mirrors our own: augment or replace the reward signal altogether. Additionally, our reparameterization by itself is not tied to the sigmoid function in any way and so, in principle, a suitable unbiased alternative could also be used in conjunction with our work.
> >
> > Minor comments: Adding the individual functions f to Table 1 would, unfortunately, push the table out of the margins; we will add such a complete table to the appendix. The title of Algorithm 1 should not be f-VIMO-sigmoid as sigmoid appears nowhere in the algorithm itself. Sigmoid is a suitable choice made for our experiments but could be replaced with another function, as discussed in Section 4.3.

---

> > > ### Comment · AnonReviewer1 · 2019-11-15
> > > **I think reward bias is a major problem in the current evaluation**
> > >
> > > I just want to note that I think that reward bias seems to be a highly plausible explanation for the presented results--well spotted!
> > >
> > > To provide some context:
> > > Kostrikov et al. (2019) showed a bug in many imitation learning implementation that stems from the fact that trajectories returned by common frameworks (include Baselines/Gym) do not include absorbing states, which prevents imitation learning algorithms from learning the reward for these states and from applying the learned reward function to these states. Instead their reward/return is implicitly set to zero. This has to be considered a bug in the implementation not a shortcoming of the derived algorithms which would require to learn the reward for all states. The effect of this bug is that methods that learn reward function that only produce negative values will always result in optimal policies that terminate the episode as quickly as possible.
> > >
> > > It seems that the author's implementation suffers from this exact bug. The bad performance of negative reward functions on "survival"-tasks is to be expected. Relating it to the buggy implementation is thus not merely some conjecture. A fair comparison needs to either learn the rewards for absorbing state or only consider environments that can not end prematurely due to task success/failure.

---

> ### Author Response · Authors · 2019-11-14
> **Swapping Distributions of Variational Lower Bound**
>
> Please see the newly added Section C.3 of the supplement addressing the reviewer's idea for swapping the positions of the distributions in the variational lower bound to the f-divergence.

---

### Official Review · AnonReviewer1 · 2019-10-23
**Official Blind Review #1**

**Rating:** 1

**Review:**

Summary: The submission performs empirical analysis on f-VIM (Ke, 2019), a method for imitation learning by f-divergence minimization. The paper especially focues on a state-only formulation akin to GAILfO (Torabi et al., 2018b). The main contributions are:
1) The paper identifies numerical proplems with the output activations of f-VIM and suggest a scheme to choose them such that the resulting rewards are bounded.
2) A regularizer that was proposed by Mescheder et al. (2018) for GANs is tested in the adversarial imitation learning setting.
3) In order to handle state-only demonstrations, the technique of GAILfO is applied to f-VIM (then denoted f-VIMO) which inputs state-nextStates instead of state-actions to the discriminator.

Contribution / Significance:
I think that the contributions of the paper are rather marginal. I do think that the choice of output activation may have large impact on the performance and it seems that the activation suggested by Ke et al. (2019) are somewhat arbitrary. However, the activations proposed in the current submission are also seem somewhat arbitrary and are not accompanied by any theoretical analysis.
2) and 3) are marginal combinations of existing work that are only insufficiently evaluated and do not seem particular effective.
Hence, I think that the current submission is of rather limited interest.

Soundness:
The "reparametrization" of f-VIM is motivated based on exploding policy gradients when using unbounded reward functions, especially when minimizing the (R)KL.
I am not convinced by this motivation, given that GAIL and AIRL (which approximatly minimizes the RKL) use unbounded reward functions and do not seem to suffer from such problems.

Evaluation:
The effect of the "reparametrization" is only evaluated for total variation. The regularization loss is only evaluated with a single fixed coefficient of 10 on all experiment. I think that a sweep over the coefficient would be mandatory, especially given that current experiments do not show a clear benefit of the regularization loss (the regularized version performs worse on roughly half of the experiments).
When learning from observations only, the submission only evaluates the proposed combination of f-VIM and GAILfO. However, it seems like it would be perfectly possible to handle state-only observations by simply making the discriminator independent of the actions, i.e. using D(s,a) = D(s). Such technique matches the marginal distributions over states and is commonly applied to GAIL, e.g. by Peng et al. [1].
It is not clear whether the reported problems of learning from observations only is really a general problem of the learning setting (as claimed in the submission) or a problem of the proposed method.

Clarity:
The paper is well written and easy to follow. Using different linestyles to distinguish the learning with regularization versus without regularization would help a lot.

Decision:
Due to the marginal contribution and the insufficient evaluation I have to recommend rejection.


Question:
I am maily interested in the authors response to my critique, especially regarding
- the choice not to compare with state-only f-VIM, and
- the motivation of the proposed output activations.


[1] Peng, Xue Bin, et al. "Variational discriminator bottleneck: Improving imitation learning, inverse rl, and gans by constraining information flow." arXiv preprint arXiv:1810.00821 (2018).

**Experience Assessment:**

I have published one or two papers in this area.

**Review Assessment: Checking Correctness Of Derivations And Theory:**

I assessed the sensibility of the derivations and theory.

**Review Assessment: Checking Correctness Of Experiments:**

I assessed the sensibility of the experiments.

**Review Assessment: Thoroughness In Paper Reading:**

I read the paper at least twice and used my best judgement in assessing the paper.

---

> ### Author Response · Authors · 2019-11-08
> **Response to Reviewer #1**
>
> “...the contributions of the paper are rather marginal” — please see the comment made to all reviewers.
>
> “...the activations proposed in the current submission are also seem somewhat arbitrary and are not accompanied by any theoretical analysis.” — We invite the reviewer to offer concrete suggestions on how to perform the suggested “theoretical analysis” of variational function activation choices when estimating f-divergences. It would no doubt serve as a useful tool for justifying the “somewhat arbitrary” choices of activation functions offered in the original f-GAN paper (Nowozin et al. 2016) that are widely used in practice. In the meantime, we assert that the choice of sigmoid activation in this work is no more arbitrary than the variety of reward hacks employed in practice throughout deep reinforcement learning to enforce stability (Henderson et al. 2018). Analogous to the arbitrary f-GAN activation choices, these heuristic selections are used because of their widespread success in rectifying practical implementation issues; in f-GANs, this is ensuring conformity to the domain of the convex conjugate. The reason why all of our figures lack plots for these f-GAN activation choices stems directly from the resulting numerical instabilities caused by exploding policy gradients. Consequently, both our choice and those made throughout the literature were done with the goal of maintaining stability during policy gradient updates where policy returns directly scale the gradient. Note that this is typically not a concern in the traditional GAN literature that employs standard end-to-end backpropagation for training both the generator and discriminator models. Our lack of theoretical analysis for reward function choice mirrors the overall lack of a theoretical understanding for reward hacks in general throughout deep reinforcement learning.
>
> “2) and 3) are marginal combinations of existing work that are only insufficiently evaluated and do not seem particular effective.” — we ask the reviewer to be precise and specify exactly which aspects of the evaluation are insufficient. As for the effectiveness of our proposed methods, we point to our Figure 4 as but a single example to note that the plots associated with traditional GAIL (top row, blue) do not strictly represent the best performance results across all environments, indicating the effectiveness of 2) and 3).
>
> “I am not convinced by this motivation, given that GAIL and AIRL (which approximatly minimizes the RKL) use unbounded reward functions and do not seem to suffer from such problems.” — the reviewer is correct that GAIL and AIRL do not seem to suffer from this problem; thus, a logical conclusion of the reviewer’s observation is that not all unbounded reward functions yield exploding policy gradients. When exploding policy gradients do occur, however, we believe the reviewer may agree that having a reparameterization to alleviate the issue would be useful. If the reviewer could provide a concrete pointer to the connection between AIRL and RKL, that would be much appreciated as we could not find such a connection in the original paper. Still, our experiments left us unable to run (R)KL experiments to completion on account of exploding policy gradients; in all experiments, the use of (R)KL-VIM(O) is always done with sigmoid rewards to rectify the instability. Note that a naive, brute-force solution to this problem does exist in the form of grid searching over all possible thresholds for gradient norm clipping of the policy in order to find one that is as large as possible without succumbing to numerical errors; solving this kind of “Goldilocks” problem would be to accept the problem rather than address it. Moreover, the computational resources needed for such a search across multiple choices of f-divergence is quite intensive and our reparameterization offers a stable optimization solution without relying on such computational inefficiency, which does constitute an important contribution.
>
> “The effect of the "reparametrization" is only evaluated for total variation” — as mentioned in the paper, both KL and RKL lead to numerical instabilities that brought all random trials to a complete halt. Since GAIL and GAIFO are already performant imitation learning algorithms on their own (and we wish to maintain fidelity to the original algorithms as baselines),  we did not examine the effect of reparameterizing with sigmoid rewards on either one.

---

> > ### Author Response · Authors · 2019-11-08
> > **Response to Reviewer #1 (continued)**
> >
> >
> > “ I think that a sweep over the coefficient would be mandatory, especially given that current experiments do not show a clear benefit of the regularization loss” — preliminary experiments varying the regularization coefficient were conducted in the Ant domain where it was seen to be ineffective and was consequently not further pursued. This is consistent with the findings of Mescheder et al. (2018) who also report insensitivity to the coefficient. We do not make any claims on the necessity of discriminator regularization to achieve strong imitation performance; our empirical results suggest that it offers potential benefit that can only be established on a per-environment and per-divergence basis.
> >
> > “ it seems like it would be perfectly possible to handle state-only observations by simply making the discriminator independent of the actions, i.e. using D(s,a) = D(s). Such technique matches the marginal distributions over states and is commonly applied to GAIL, e.g. by Peng et al. [1].” — while restricting the discriminator focus to states only (instead of state transitions as done in this work and GAIFO) is possible, notice that there are cases where such a state-marginal matching imitation algorithm can dramatically fail. Consider a simple example: a tabular MDP of N states organized in a ring with transitions only between the two adjacent neighbors of each state; an expert policy that moves clockwise and an imitation policy that moves anticlockwise will yield identical state marginal distributions while clearly failing the imitation task.
> >
> > “I am maily interested in the authors response to my critique, especially regarding - the choice not to compare with state-only f-VIM, and - the motivation of the proposed output activations.” — please see our response just above for the first question and the second response for the second question.

---

> > > ### Comment · AnonReviewer1 · 2019-11-15
> > > **Some replies to your reply**
> > >
> > > "We invite the reviewer to offer concrete suggestions on how to perform the suggested “theoretical analysis” of variational function activation choices when estimating f-divergences."
> > > - I can't! I just wanted to point out that the sigmoid activation is just as arbitrary as the activations proposed by Nowozin et al. (2016).
> > >
> > > "we ask the reviewer to be precise and specify exactly which aspects of the evaluation are insufficient."
> > > - I believe I did this already in my original rebuttal:
> > > Regarding the regularization, I do not think that its effect can be properly evaluated when evaluating it only for a single coefficient (we apparently disagree here).
> > > Regarding the state-nextState-Discriminator, I think that comparing it with a state-only-discriminator would be a useful comparison. Apparently we also disagree here. Yes, we can construct tasks where state-only matching fails. But, how does this relate to the experiments in the paper? Also note that we can also construct tasks where matching state-transitions fails, e.g. tasks where different actions lead to the same state but have different "energy" costs.
> > >
> > > "reference for AIRL minimizing the reverse KL"
> > > - I think you referred to that paper yourself. It is:
> > > Ghasemipour et al."A Divergence Minimization Perspective on Imitation Learning Methods". 2019.
> > > See Section 4.1 in https://arxiv.org/pdf/1911.02256.pdf

---

### Official Review · AnonReviewer3 · 2019-10-23
**Official Blind Review #3**

**Rating:** 1

**Review:**

This paper proposes the application of the f-VIM framework (Ke et. al., 2019) to the problem of imitation learning from observations (no expert actions). The authors first identify a potential source of numerical instability in the application of f-VIM to imitation learning – the rewards for the policy-gradient RL are given by a combination of a convex conjugate and an activation function. To alleviate this, f-VIM is reparameterized by curating the activation using conjugate inverse (Equation 8), yielding a potentially more stable reward for deep-RL.

I have the following concerns about the paper:

1.	Lack of novelty – Although I appreciate the reparameterization applied to f-VIM to make it potentially more stable for imitation learning in large state- and action-spaces, I don’t think that by itself meets the bar for ICLR. Algorithm 1 is basically the GAILFO algorithm (Torabi et al. 2018) written in the f-Vim framework, with the proposed reparameterization. The discriminator regularization (Section 4.4) has been used before.

2.	Experiments – Figure 2 shows the improvement with TV when using the reparameterization, and the authors mention in text about the difficulty with KL and reverse-KL. What about the JS divergence (GAIL)? Does reparameterization help or affect that?

3.	In Figure 3, is GAIL from the original paper, or does it use the sigmoid rewards? Figure 3 does not offer any evidence that the proposed methods in the paper lead to algorithms that should be preferred over the current state-of-the-art in imitation learning with divergence minimization such GAIL and WAIL.

Minor comment:
In Table 1: GAN is not a divergence. Please use Jensen-Shannon, with the corresponding tweaks to the columns.

**Experience Assessment:**

I have published one or two papers in this area.

**Review Assessment: Checking Correctness Of Derivations And Theory:**

I assessed the sensibility of the derivations and theory.

**Review Assessment: Checking Correctness Of Experiments:**

I assessed the sensibility of the experiments.

**Review Assessment: Thoroughness In Paper Reading:**

I read the paper thoroughly.

---

> ### Author Response · Authors · 2019-11-08
> **Response to Reviewer #3**
>
> “Lack of novelty – Although I appreciate the reparameterization applied to f-VIM to make it potentially more stable for imitation learning in large state- and action-spaces, I don’t think that by itself meets the bar for ICLR” — The statement that our paper does not meet “the bar for ICLR” is completely opaque; we would appreciate an explicit description from the reviewer that characterizes the current gap between our paper and “the bar for ICLR.” Please see the response made to all reviewers concerning the novelty of our approach.
>
> “What about the JS divergence (GAIL)? Does reparameterization help or affect that?” — We view our reparameterization of the f-VIM/VIMO objective as a remedy for instability, without which, an IL or ILfO algorithm under a certain choice of f-divergence may trivially fail to produce meaningful imitation policies. Since GAIL and GAIFO are clearly already performant imitation algorithms on their own (and we wish to maintain fidelity to the original algorithms as baselines),  we did not examine the effect of sigmoid rewards on either one.
>
> “In Figure 3, is GAIL from the original paper, or does it use the sigmoid rewards?” — As previously mentioned, our use of sigmoid rewards is only done to remedy what would otherwise be an unstable imitation algorithm. As the original GAIL and GAIFO algorithms do not suffer from such instabilities, any reported results for either algorithm adheres to their original papers; that is, we never use sigmoid rewards with either GAIL or GAIFO. We will make this clearer in the paper.
>
> “Figure 3 does not offer any evidence that the proposed methods in the paper lead to algorithms that should be preferred over the current state-of-the-art in imitation learning with divergence minimization such GAIL and WAIL.” — please see our response to this comment in our reply to all reviewers.
>
> Minor comments: We agree with the reviewer that GAN is not itself a divergence between probability distributions. However, the divergence optimized by GANs is also not exactly the Jensen-Shannon divergence (something we note explicitly as a footnote in our extended review of prior work in the appendix and detailed more in Nowozin et al. 2016). Our presentation is consistent with that of Nowozin et al. (2016) while maintaining correctness. We will certainly add the Jensen-Shannon divergence to our tables in order to highlight this fact explicitly.

---

### Author Response · Authors · 2019-11-08
**Comments for all Reviewers**

We thank the reviewers for providing feedback on our submission.

There seems to be a bit of confusion amongst some of the reviewers concerning the core contribution and central hypothesis of this paper; our apologies that these did not come across well and we plan to use the discussion during the rebuttal period to improve the overall clarity of our paper.

Concretely, Reviewer #2 mentions that we evaluate “existing methods (f-VIM and f-VIMO) with difference choices of divergence.” While f-VIM (Ke et al. 2019) is an algorithm from prior work, f-VIMO is a novel contribution of this paper for which the only existing counterpart is GAIFO (Torabi et al. 2018). Moreover, the central hypothesis investigated in this work is that the exploration of alternative f-divergences, rather than the standard Jensen-Shannon divergence employed by GANs, may yield benefits when learning imitation policies from observations only (without the provision of expert action labels). Reviewer #3 mentions that “Figure 3 does not offer any evidence that the proposed methods in the paper lead to algorithms that should be preferred over the current state-of-the-art in imitation learning with divergence minimization such GAIL and WAIL.” This is true and in line with the core goal of this paper: to assess the potential for superior *imitation-from-observation* algorithms, as opposed to the traditional imitation learning setting. The inclusion of imitation learning results in our plots is to act as an intuitive upper bound on what should be achievable relative to the imitation from observation setting.

Both Reviewers #1 and #3 comment on the lack of novelty in our paper, claiming that “the contributions of the paper are rather marginal.” We ask the reviewers to keep in mind that the story told in this paper transitions from the GAIFO algorithm of Torabi et al., (2018) to arbitrary f-divergences through the f-VIMO algorithm. This is analogous to prior transitions from GANs (Goodfellow et al., 2014) to f-GANs (Nowozin et al., 2016) as well as from GAIL (Ho & Ermon, 2016) to f-VIM (Ke et al., 2019). We assert that the existence of these past parallelisms in the literature do not diminish the novelty of our work or alter the distinctness of our problem setting, namely the (strictly harder) imitation learning from observation (ILfO) problem. In addition to drawing the parallel, we resolve practical issues that arise when deploying these algorithms to achieve reasonable imitation policies through stable policy-gradient optimization. Additionally, while the discriminator regularization (Mescheder et al. 2018) has been used before in the traditional GAN setting, it has, to the best of our knowledge, never been assessed in the context of imitation learning or ILfO settings.

---

### Decision · Program_Chairs · 2019-12-19

**Decision:**

Reject

**Comment:**

The submission performs empirical analysis on f-VIM (Ke, 2019), a method for imitation learning by f-divergence minimization. The paper especially focues on a state-only formulation akin to GAILfO (Torabi et al., 2018b). The main contributions are:
1) The paper identifies numerical proplems with the output activations of f-VIM and suggest a scheme to choose them such that the resulting rewards are bounded.
2) A regularizer that was proposed by Mescheder et al. (2018) for GANs is tested in the adversarial imitation learning setting.
3) In order to handle state-only demonstrations, the technique of GAILfO is applied to f-VIM (then denoted f-VIMO) which inputs state-nextStates instead of state-actions to the discriminator.

The reviewers found the submitted paper hard to follow, which suggests a revision might make more apparent the author's contributions in later submissions of this work.